Crucial lncRNAs associated with adipocyte differentiation from human adipose-derived stem cells based on co-expression and ceRNA network analyses

Chen Kana 1
Xie Shujie 2
Jin Wujun 1 wujunjin2019@163.com
1 Department of Plastic Surgery, Hwa Mei Hospital, University of Chinese Academy of Sciences , Ningbo, Zhejiang , China
2 Department of Hepatobiliary Surgery, Hwa Mei Hospital, University of Chinese Academy of Sciences , Ningbo, Zhejiang , China
Nakai Kenta
Electronic publication date: 2019 Sep 6
Publication date: 2019
Volume: 7
Electronic Location ID: e7544
Received 2019 May 15; Accepted 2019 Jul 24
Copyright: © 2019 Chen et al.
Copyright year: 2019
Copyright holder: Chen et al.
License: This is an open access article distributed under the terms of the Creative Commons Attribution License, which permits unrestricted use, distribution, reproduction and adaptation in any medium and for any purpose provided that it is properly attributed. For attribution, the original author(s), title, publication source (PeerJ) and either DOI or URL of the article must be cited.
License URL: https://creativecommons.org/licenses/by/4.0/

Keywords: Human adipose tissue-derived stromal stem cells, Adipogenic differentiation, ceRNA, lncRNA, miRNA, Co-expression

Funding: The authors received no funding for this work.

==============================
Background

Injection of adipose-derived stem cells (ASCs) is a promising treatment for facial contour deformities. However, its treatment mechanisms remain largely unknown. The study aimed to explain the molecular mechanisms of adipogenic differentiation from ASCs based on the roles of long noncoding RNAs (lncRNAs).

Methods

Datasets of mRNA–lncRNA (GSE113253) and miRNA (GSE72429) expression profiling were collected from Gene Expression Omnibus database. The differentially expressed genes (DEGs), lncRNAs (DELs) and miRNAs (DEMs) between undifferentiated and adipocyte differentiated human ASCs were identified using the Linear Models for Microarray Data method. DELs related co-expression and competing endogenous RNA (ceRNA) networks were constructed. Protein–protein interaction (PPI) analysis was performed to screen crucial target genes.

Results

A total of 748 DEGs, 17 DELs and 51 DEMs were identified. A total of 13 DELs and 279 DEGs with Pearson correlation coefficients > 0.9 and p-value < 0.01 were selected to construct the co-expression network. A total of 151 interaction pairs among 112 nodes (10 DEMs; eight DELs; 94 DEGs) were obtained to construct the ceRNA network. By comparing the lncRNAs and mRNAs in two networks, five lncRNAs (SNHG9, LINC02202, UBAC2-AS1, PTCSC3 and myocardial infarction associated transcript (MIAT)) and 32 genes (i.e., such as phosphoinositide-3-kinase regulatory subunit 1 (PIK3R1), protein tyrosine phosphatase receptor type B (PTPRB)) were found to be shared. PPI analysis demonstrated PIK3R1 , forkhead box O1 (FOXO1; a transcription factor) and estrogen receptor 1 (ESR1) were hub genes, which could be regulated by the miRNAs that interacted with the above five lncRNAs, such as LINC02202-miR-136-5p-PIK3R1, LINC02202-miR-381-3p-FOXO1 and MIAT-miR-18a-5p-ESR1. LINC02202 also could directly co-express with PIK3R1. Furthermore, PTPRB was predicted to be modulated by co-expression with LINC01119.

Conclusion

MIAT, LINC02202 and LINC01119 may be potentially important, new lncRNAs associated with adipogenic differentiation of ASCs. They may be involved in adipogenesis by acting as a ceRNA or co-expressing with their targets.

Introduction

Autologous adipose tissue grafting has been a widely accepted surgical tool for anti-aging cosmetics (Charles-De-Sá et al., 2015) and reconstructive restoration of various congenital or acquired facial soft tissue deformities (Bashir et al., 2018). However, conventional fat grafting procedure needs to be repeated multiple times to achieve satisfactory results (Bashir et al., 2018), which may be associated with the low graft survival rate and poor revascularization (Ma et al., 2015). To overcome these two limitations, recent scholars propose to combine with additional autologous adipose-derived stem cells (ASCs) which have the ability to differentiate into mature adipocytes to supplement apoptotic cells and secrete angiogenic growth factors to enhance angiogenesis (Bashir et al., 2018; Kotaro et al., 2008; Philips, Marra & Rubin, 2014). The clinical trials also confirm that supplementation of ASCs to adipose grafts is superior to conventional lipoinjection for facial recontouring (Bashir et al., 2018; Kotaro et al., 2008). Nevertheless, the use of autologous ASCs has not been FDA-approved. This may be because there still remains a huge gap in understanding the potential mechanisms of ASCs for adipocyte differentiation.

Increasing evidence has suggested long noncoding RNAs (lncRNAs), a class of noncoding RNAs more than 200 nucleotides, play crucial roles in adipogenesis for ASCs. For example, Nuermaimaiti et al. (2018) demonstrated that knockdown of HOXA11-AS1 inhibited adipocyte differentiation, leading to suppression of adipogenic-related gene transcription, as well as decreased lipid accumulation in ASCs. Huang et al. (2017) observed knockdown of MIR31HG inhibited adipocyte differentiation, whereas overexpression of MIR31HG promoted adipogenesis in vitro and in vivo. MEG3 was also found to be downregulated during adipogenesis of ASCs. Functional analysis showed that knockdown of MEG3 promoted adipogenic differentiation of ASCs (Li et al., 2017). Furthermore, current research shows lncRNAs, on one hand, functions as microRNA (miRNAs) sponges to bind the miRNA response elements and regulate miRNA-mediated gene silencing (i.e., competing endogenous RNA (ceRNA) hypothesis); and, on the other hand, directly influences their neighboring genes expression by chromatin remodeling or transcriptional control (co-expression model) (Huang et al., 2016; Li, Ao & Wu, 2017). These theories have also been reported in ASCs. Li, Ao & Wu (2017) proved downregulated MEG3 may be insufficient to sponge miR-140-5p and lead to its upregulation during adipogenesis in ASCs. Huang et al. (2017) revealed inhibition of MIR31HG reduced the enrichment of active histone markers, histone H3 lysine 4 trimethylation and acetylation in the promoter of fatty acid binding protein 4, resulting in suppression of its expression and adipogenesis. However, the adipogenic differentiation related lncRNAs and its mechanisms of ASCs remains rarely reported.

The present study aimed to identify crucial lncRNAs involved in adipocyte differentiation of ASCs by constructing lncRNA–miRNA–mRNA ceRNA network and lncRNA–mRNA co-expression network using high throughput analysis data. Our findings might offer greater insights into the molecular mechanisms of adipocyte differentiation from ASCs and provide potentially new targets for inducing adipogenesis.

Materials and Methods

Collection of microarray data

GSE113253 (Rauch et al., 2019) and GSE72429 datasets (Supplemental Information 1) were downloaded from the Gene Expression Omnibus (GEO) database (http://www.ncbi.nlm.nih.gov/geo/). GSE113253 dataset applied the high throughput sequencing methodology to simultaneously detect the lncRNA and mRNA expression profiles in two repeats of undifferentiated human ASCs and 10 repeats of adipogenic differentiation cells using an Illumina HiSeq 1500 instrument, which was submitted to GEO on April 17, 2018. GSE72429 dataset analyzed the miRNA expression profile in four undifferentiated human ASCs and two adipogenic differentiation cells using an Agilent-031181 Unrestricted_Human_miRNA_V16.0_Microarray (miRBase release 16.0 miRNA ID version), which was submitted to GEO on August 27, 2015.

Differential expression analysis

The normalized series matrix files of each dataset were downloaded from GEO. Following re-annotation according to corresponding platform (GPL18460), the expression values of the lncRNAs and mRNAs in GSE113253 were obtained. The differentially expressed genes (DEGs), lncRNAs (DELs) and miRNAs (DEMs) were identified using the Linear Models for Microarray Data method software (version 3.34.0; Ritchie et al., 2015). p-Value was adjusted by using Benjamini–Hochberg method to avoid false positives. The heatmap was constructed to present the expression difference of DEGs, DELs and DEMs in different samples using the pheatmap package (version: 1.0.8; Kolde, 2019) based on Euclidean distance.

Co-expression network between lncRNA and mRNA

The co-expression network was constructed based on the correlation analysis between DELs and DEGs. Pearson correlation coefficients were calculated using the Weighted Gene Correlation Network Analysis (Langfelder & Horvath, 2016) algorithm to assess the correlation. Only the co-expressed pairs with absolute value of Pearson correlation coefficients ≥ 0.9 and p < 0.01 were selected to draw the network using Cytoscape (version 3.4; Shannon et al., 2001–2008; Kohl, Wiese & Warscheid, 2011).

CeRNA regulatory network among DELs, DEMs and DEGs

The DEMs related target genes were predicted using the miRwalk database (version 2.0; Dweep & Gretz, 2015a, 2015b) which provides 12 prediction algorithms (miRWalk, MicroT4, miRanda, miRBridge, miRDB, miRMap, miRNAMap, PICTAR2, PITA, RNA22, RNAhybrid, Targetscan). Only the miRNA-target gene interaction pairs that were predicted in at least eight databases were used. The target genes were then overlapped with the DEGs to screen negatively correlated DEM–DEG interaction pairs. The miRcode (http://www.mircode.org/) (Jeggari, Marks & Larsson, 2012), starBase (version 2.0; Yang, 2010–2013; Li et al., 2014) and DIANA-LncBase (version 2.0; Paraskevopoulou et al., 2019; Paraskevopoulou et al., 2013) databases were used to predict the interaction relationship between DELs and DEMs. The negatively correlated DEL–DEM interaction pairs were left for further analysis. The DEL–DEM and DEM–DEG interactors were integrated to construct the ceRNA network, which was visualized using Cytoscape.

Protein–protein interaction network

Protein–protein interaction (PPI) data of DEGs in the ceRNA network was collected from Search Tool for the Retrieval of Interacting Genes (STRING; version 10.0; Szklarczyk et al., 2019) database (Szklarczyk et al., 2015). Only interactions with combined score >0.4 were selected to construct the PPI network. Several topological features of the nodes (protein) in the PPI network were calculated using the CytoNCA plugin in cytoscape software (Tang, Li & Wang, 2014; Tang et al., 2015) to screen hub genes, including degree, eigenvector, betweenness and closeness centrality. Furthermore, transcription factors were predicted using iRegulon (Janky et al., 2014) in Cytoscape and then integrated to the PPI network.

Function enrichment analysis

Gene ontology (GO) and Kyoto Encyclopedia of Genes and Genomes (KEGG) pathway enrichment analyses were performed using the Database for Annotation, Visualization and Integrated Discovery online tool (version 6.8; http://david.abcc.ncifcrf.gov) (Huang, Sherman & Lempicki, 2009) to reveal the function of DEGs. p < 0.05 was set as the cut-off value.

Results

Differential expression analysis

Due to the fact that fewer DEGs, DELs and DEMs were identified if adjusted p-value was defined as the statistical threshold; therefore, genes, lncRNAs and miRNAs were believed to be differentially expressed in this study when their |log2fold change (FC)| was more than 1 and p-value was less than 0.05. Based on these given thresholds, a total of 748 protein-coding genes (360, upregulated; 388, downregulated) (Table 1; Supplemental Information 2) and 17 lncRNAs (nine upregulated; eight downregulated) (Table 1; Supplemental Information 2) were found to be differentially expressed in adipogenic differentiation cells compared with undifferentiated cells in GSE113253 dataset. Among them, 121 DEGs (such as forkhead box O1 (FOXO1), protein tyrosine phosphatase receptor type B (PTPRB)) and two DELs (SH3RF3-AS1, LINC01119) had adjusted p-value < 0.05, indicating they were especially crucial for adipogenic differentiation. A total of 51 miRNAs (Table 1; Supplemental Information 2) were identified to be significantly differentially expressed in GSE72429 within the p < 0.05 and |log2FC| > 1 criteria. Among them, 20 DEMs (particularly, miR-663 and miR-3607-3p, with adjusted p-value < 0.05) were upregulated and 31 DEMs (particularly, miR-150*, miR-4271, miR-371-5p and miR-134, with adjusted p-value < 0.05) were downregulated. Additionally, hierarchical clustering of DEGs (Fig. 1A), DELs (Fig. 1B) and DEMs (Fig. 1C) expression levels indicated the differentiated samples could be well distinguished from the undifferentiated samples.

Table 1 Differentially expressed genes, lncRNAs and miRNAs.

	logFC	p-Value		logFC	p-Value		logFC	p-Value	
CRLF1	5.31	1.55E-10*	SH3RF3-AS1	4.00	8.45E-06*	miR-663	6.22	2.53E-06*	
ZBTB16	6.71	2.90E-10*	LINC01554	2.06	1.19E-02	miR-3607-3p	5.55	3.91E-06*	
COMP	6.30	5.07E-10*	SNHG9	2.04	1.45E-02	miR-455-3p	2.93	2.91E-04	
FOXO1	4.98	3.85E-09*	LINC01914	2.35	1.74E-02	miR-455-5p	5.68	6.45E-03	
LMO3	4.92	5.15E-09*	C18orf65	1.61	2.40E-02	miR-30c	1.45	7.52E-03	
KLF15	5.19	6.31E-09*	LINC02202	1.65	4.06E-02	miR-181b	1.45	1.26E-02	
MT1G	4.71	1.68E-08*	UBAC2-AS1	1.69	4.17E-02	miR-92a	1.33	1.33E-02	
NEFL	5.50	3.12E-08*	LOH12CR2	1.86	4.55E-02	miR-609	2.43	2.97E-02	
NRCAM	4.84	5.19E-08*	OSER1-DT	1.92	4.99E-02	miR-339-3p	2.48	2.99E-02	
PCSK1	4.32	1.05E-07*	LINC01119	−3.77	1.80E-04*	miR-887	2.65	3.07E-02	
FRAS1	6.29	1.14E-07*	SERPINB9P1	−2.35	4.55E-03	miR-124	2.70	3.10E-02	
PDK4	5.95	1.50E-07*	MIAT	−3.24	5.06E-03	miR-3653	1.01	3.17E-02	
PER1	3.90	3.26E-07*	LINC00601	−1.85	1.98E-02	miR-652	2.90	3.20E-02	
IL18R1	4.60	4.67E-07*	LINC00211	−1.48	3.31E-02	miR-769-5p	2.91	3.21E-02	
MT1X	3.60	4.76E-07*	PTCSC3	−1.72	3.81E-02	miR-18a	3.08	3.30E-02	
MT1M	3.82	4.86E-07*	CYTOR	−1.49	4.63E-02	miR-1290	3.31	3.42E-02	
PDE4D	4.06	1.03E-06*	LINC00865	−1.53	4.85E-02	miR-1973	1.15	3.45E-02	
SERPINA3	4.91	1.61E-06*	SH3RF3-AS1	4.00	8.45E-06*	miR-30a*	1.84	3.87E-02	
RASD1	4.46	1.68E-06*				miR-132	1.23	3.87E-02	
IL1RL1	5.91	1.71E-06*				miR-K12-5*	2.93	4.46E-02	
GALNT15	4.46	2.24E-06*				miR-150*	−6.28	3.37E-07*	
FKBP5	3.42	2.37E-06*				miR-4271	−6.29	1.43E-06*	
ELOVL3	3.94	2.95E-06*				miR-371-5p	−6.51	1.79E-06*	
HSD11B1	3.63	3.39E-06*				miR-134	−6.39	3.26E-06*	
PIK3R1	2.42	5.88E-04				miR-146b-5p	−5.32	4.43E-04	
ARNT2	−5.34	3.01E-08*				miR-136	−2.12	1.12E-03	
FGF9	−4.24	2.78E-07*				miR-199b-5p	−2.81	4.17E-03	
IL6	−5.82	2.90E-07*				miR-29b	−1.80	5.88E-03	
OXTR	−5.64	2.06E-06*				miR-376b	−4.36	9.46E-03	
RTKN2	−4.16	2.83E-06*				miR-130b	−1.33	1.20E-02	
PTPRB	−4.63	4.43E-06*				miR-218	−3.94	1.22E-02	
SHROOM3	−3.37	7.27E-06*				miR-154*	−4.42	1.25E-02	
RGS4	−3.28	8.10E-06*				miR-381	−1.11	1.31E-02	
ARHGEF28	−3.39	1.08E-05*				miR-377	−1.23	1.45E-02	
GPR68	−3.67	1.46E-05*				miR-503	−1.85	1.59E-02	
VCAM1	−4.30	1.59E-05*				miR-337-5p	−1.04	2.13E-02	
ATP8B1	−3.51	1.80E-05*				miR-3132	−1.24	2.18E-02	
CNIH3	−3.42	2.61E-05*				miR-362-3p	−3.63	2.18E-02	
ZSWIM4	−3.17	2.67E-05*				miR-3659	−1.28	2.23E-02	
EPHA2	−3.36	3.35E-05*				miR-H6	−1.49	2.31E-02	
CDCP1	−4.48	3.93E-05*				miR-135a*	−2.56	2.37E-02	
FRMD5	−3.14	4.35E-05*				miR-29b-1*	−1.79	2.61E-02	
NR3C2	−2.88	4.40E-05*				miR-376c	−1.09	2.64E-02	
GREM2	−3.48	4.91E-05*				miR-193a-3p	−1.35	2.79E-02	
CEMIP	−4.84	4.92E-05*				miR-140-3p	−1.07	2.84E-02	
BIRC3	−3.21	8.05E-05*				miR-642b	−3.77	3.04E-02	
RBM24	−3.28	9.30E-05*				miR-125a-3p	−1.20	3.16E-02	
KY	−3.20	1.08E-04*				miR-140-5p	−1.26	3.51E-02	
NUAK2	−2.93	1.21E-04*				miR-718	−2.64	4.41E-02	
FGF1	−4.52	1.27E-04*				miR-299-3p	−4.10	4.83E-02	
ESR1	−1.911	1.48E-02				miR-376a*	−4.02	4.99E-02	
Note:

All the differentially expressed miRNAs and lncRNAs were shown, but only top 25 upregulated and downregulated mRNAs as well as crucial genes were displayed.

FC, fold change.

p-Value with asterisk indicated their adjusted p-value were also less than 0.05.

Figure 1 Hierarchical clustering and heat map analysis of differentially expressed (A) genes, (B) long non-coding RNAs and (C) microRNAs.

Red, high expression; light blue, low expression.

Construction of co-expression and ceRNA networks

A total of 13 DELs and 279 DEGs with Pearson correlation coefficients > 0.9 and p-value < 0.01 were selected to construct the lncRNA–mRNA co-expression network, which contained 440 positive connections (Fig. 2; Supplemental Information 3).

Figure 2 Co-expression network between differentially expressed long non-coding RNAs and genes.

(A) Downregulated lncRNA–mRNA co-expression (blue); (B) upregulated lncRNA–mRNA co-expression (red). Circular, differentially expressed genes; rhombus, differentially expressed long non-coding RNAs.

Based on at least eight database analyses in miRwalk 2.0 and negatively correlated principles, a total of 79 downregulated DEGs were predicted to be regulated by eight upregulated DEMs, while 128 upregulated DEGs were predicted to be regulated by 32 downregulated DEMs. Using the starBase database, 355 miRNAs were predicted to interact with 25 DELs; using the miRcode database, 192 miRNAs were predicted to interact with eight DELs; using the DIANA-LncBase database, 1,343 miRNAs were predicted to interact with 15 DELs. After overlapping the DEMs that interacted with DELs and DEMs that regulated DEGs, 151 interaction pairs among 112 nodes (10 DEMs, four upregulated and six downregulated; eight DELs, four upregulated and four downregulated; 94 DEGs, 46 upregulated and 48 downregulated) were obtained, which were used for constructing the ceRNA network (Fig. 3; Supplemental Information 4).

Figure 3 Competing endogenous RNA network (ceRNA) among differentially expressed long non-coding RNAs, microRNAs and genes.

(A) Downregulated ceRNA axes according to the expression of miRNAs; (B) upregulated ceRNA axes according to the expression of miRNAs. Red, upregulated; blue, downregulated. Circular, differentially expressed genes; rhombus, differentially expressed long non-coding RNAs; triangle, microRNAs.

PPI network

Protein–protein interaction pairs were predicted for the 94 DEGs in the ceRNA network using the STRING database, which resulted in 80 interaction relationship pairs that were screened between 58 nodes (24 upregulated and 34 downregulated) (Fig. 4). Phosphoinositide-3-kinase regulatory subunit 1 (PIK3R1), FYN proto-oncogene, Src family tyrosine kinase and estrogen receptor 1 (ESR1) were considered as hub genes in the PPI network because they ranked the top 10 in all four topological features (Table 2). In addition, FOXO1, which was included in the PPI network, was predicted as a differentially expressed transcription factor to regulate the other target genes in the PPI network using IRegulon plug-in (Fig. 4), indicating FOXO1 was also a hub gene.

Figure 4 Protein–protein interaction network.

Red, upregulated; blue, downregulated. Oval, differentially expressed genes; hexagon, differentially expressed transcription factor.

Table 2 Hub genes in the protein-protein network screened by topological features.

Gene	Degree		Betweenness	Closeness	Eigenvector	
FYN	12	FYN	983.10	FYN	0.096	FYN	0.42	
PIK3R1	10	PIK3R1	716.77	PIK3R1	0.096	PIK3R1	0.39	
ESR1	8	ESR1	658.76	KLF5	0.095	NTF3	0.33	
NTF3	7	GATA6	543.73	ESR1	0.095	EPHA2	0.32	
EPHA2	7	KLF5	529.64	EPHA2	0.093	NRG1	0.30	
WNT5A	7	SOX5	438	NRG1	0.093	EPHA4	0.24	
NRG1	6	WNT5A	405.05	FOXO1	0.092	FGF1	0.24	
FOXO1	6	PLXNA2	352	NTF3	0.092	MBP	0.22	
EPHA4	5	CELF2	274	ITGA2	0.092	MAP2	0.18	
MBP	5	GDF5	270	GATA6	0.091	ESR1	0.15	

Function analysis showed eight significant KEGG pathways were enriched, including hsa04015:Rap1 signaling pathway (PIK3R1), hsa05200:Pathways in cancer (PIK3R1, FOXO1), hsa05205:Proteoglycans in cancer (ESR1, PIK3R1), hsa04014:Ras signaling pathway (PIK3R1), hsa05218:Melanoma (PIK3R1) and hsa04520:Adherens junction (PTPRB) (Table 3).

Table 3 KEGG pathway enrichment for the genes in the PPI network.

Term	p-Value	Genes	
hsa04015:Rap1 signaling pathway	2.25E-03	RASSF5, ID1, PDGFD, FGF1, PIK3R1, EPHA2	
hsa05200:Pathways in cancer	7.28E-03	WNT5A, EDNRB, RASSF5, FOXO1, ITGA2, FGF1, PIK3R1	
hsa05205:Proteoglycans in cancer	1.18E-02	WNT5A, ESR1, ITGA2, PIK3R1, PLAU	
hsa04014:Ras signaling pathway	1.78E-02	RASSF5, PDGFD, FGF1, PIK3R1, EPHA2	
hsa04360:Axon guidance	1.90E-02	EPHA4, FYN, PLXNA2, EPHA2	
hsa04390:Hippo signaling pathway	2.97E-02	WNT5A, ID1, GDF5, FGF1	
hsa04520:Adherens junction	4.03E-02	PTPRB, WASF3, FYN	
hsa05218:Melanoma	4.03E-02	PDGFD, FGF1, PIK3R1	

In addition, 79 GO biological process terms were also enriched, such as GO:0042981—regulation of apoptotic process (ESR1), GO:0045893—positive regulation of transcription, DNA-templated (ESR1, FOXO1), GO:0043066—negative regulation of apoptotic process (FOXO1, PIK3R1), GO:0014066—regulation of phosphatidylinositol 3-kinase signaling (PIK3R1), GO:0048146—positive regulation of fibroblast proliferation (ESR1), GO:0001525—angiogenesis (PTPRB) and GO:0001678—cellular glucose homeostasis (FOXO1, PIK3R1) (Table 4; Supplemental Information 5).

Table 4 GO biological process term enrichment for the genes in the PPI network.

Term	p-Value	Genes	
GO:0042981—regulation of apoptotic process	6.04E-05	RASSF5, TNFRSF11B, DUSP1, NTF3, FYN, GDF5, ESR1	
GO:0032148—activation of protein kinase B activity	7.76E-05	WNT5A, NTF3, FGF1, NRG1	
GO:0007596—blood coagulation	3.06E-04	PRKAR2B, FYN, GATA6, ITGA2, PDGFD, PLAU	
GO:0071560—cellular response to transforming growth factor beta stimulus	5.22E-04	WNT5A, FYN, SOX5, PDGFD	
GO:0043066—negative regulation of apoptotic process	6.03E-04	WNT5A, EDNRB, DUSP1, RPS6KA1, ID1, GATA6, FOXO1, PIK3R1	
GO:0045944—positive regulation of transcription from RNA polymerase II promoter	1.01E-03	KLF5, WNT5A, RPS6KA1, GATA6, HIPK2, TFEB, ESR1, FOXO1, FGF1, NRG1, PIK3R1	
GO:0045893—positive regulation of transcription, DNA-templated	1.25E-03	KLF5, WNT5A, GATA6, HIPK2, TFEB, ESR1, FOXO1, LGR4	
GO:0018108—peptidyl-tyrosine phosphorylation	1.47E-03	EPHA4, FYN, FGF1, NRG1, EPHA2	
GO:0014066—regulation of phosphatidylinositol 3-kinase signaling	2.02E-03	FYN, FGF1, NRG1, PIK3R1	
GO:0030335—positive regulation of cell migration	2.88E-03	NTF3, PDGFD, FGF1, PIK3R1, PLAU	
GO:0046854—phosphatidylinositol phosphorylation	3.43E-03	FYN, FGF1, NRG1, PIK3R1	
GO:0030182—neuron differentiation	3.53E-03	WNT5A, ID1, HIPK2, EPHA2	
GO:0008284—positive regulation of cell proliferation	3.70E-03	KLF5, EDNRB, NTF3, HIPK2, PDGFD, FGF1, NRG1	
GO:0048015—phosphatidylinositol-mediated signaling	4.80E-03	FYN, FGF1, NRG1, PIK3R1	
GO:0000187—activation of MAPK activity	4.93E-03	WNT5A, NTF3, FGF1, NRG1	
GO:0045892—negative regulation of transcription, DNA-templated	5.16E-03	WNT5A, ID1, GATA6, FOXO1, NRG1, ZBTB18, LGR4	
GO:0045766—positive regulation of angiogenesis	6.02E-03	WNT5A, GATA6, HIPK2, FGF1	
GO:0000122—negative regulation of transcription from RNA polymerase II promoter	7.94E-03	KLF5, EDNRB, ID1, GATA6, HIPK2, ESR1, FOXO1, ZBTB18	
GO:0043524—negative regulation of neuron apoptotic process	8.79E-03	NTF3, FYN, GDF5, HIPK2	
GO:0035556—intracellular signal transduction	9.37E-03	PRKAR2B, RASSF5, DUSP1, RPS6KA1, FYN, NRG1	
GO:0045213—neurotransmitter receptor metabolic process	9.62E-03	DMD, NRG1	
GO:0060750—epithelial cell proliferation involved in mammary gland duct elongation	1.28E-02	WNT5A, ESR1	
GO:0048146—positive regulation of fibroblast proliferation	1.31E-02	WNT5A, ESR1, PDGFD	
GO:0043406—positive regulation of MAP kinase activity	1.54E-02	PDE5A, PDGFD, FGF1	
GO:0043627—response to estrogen	1.86E-02	TNFRSF11B, GATA6, ESR1	
GO:0014068—positive regulation of phosphatidylinositol 3-kinase signaling	1.86E-02	FYN, PDGFD, NRG1	
GO:0048841—regulation of axon extension involved in axon guidance	2.23E-02	PLXNA4, PLXNA2	
GO:0008366—axon ensheathment	2.23E-02	NRG1, MBP	
GO:0050966—detection of mechanical stimulus involved in sensory perception of pain	2.54E-02	FYN, ITGA2	
GO:0008286—insulin receptor signaling pathway	2.61E-02	PDK4, FOXO1, PIK3R1	
GO:0090630—activation of GTPase activity	2.67E-02	WNT5A, NTF3, EPHA2	
GO:0060068—vagina development	2.89E-02	WNT5A, ESR1	
GO:0021785—branchiomotor neuron axon guidance	2.86E-02	PLXNA4, PLXNA2	
GO:0007165—signal transduction	3.11E-02	TNFRSF11B, NTF3, RPS6KA1, PDE5A, NR3C2, ESR1, FGF1, PIK3R1, PLAU	
GO:0048013—ephrin receptor signaling pathway	3.12E-02	EPHA4, FYN, EPHA2	
GO:0031643—positive regulation of myelination	3.17E-02	WASF3, NRG1	
GO:0010976—positive regulation of neuron projection development	3.33E-02	WNT5A, FYN, DMD	
GO:1901653—cellular response to peptide	3.48E-02	KLF5, ID1	
GO:0046849—bone remodeling	3.48E-02	LGR4, EPHA2	
GO:0033628—regulation of cell adhesion mediated by integrin	3.48E-02	EPHA2, PLAU	
GO:0001525—angiogenesis	3.49E-02	KLF5, PTPRB, ID1, FGF1	
GO:0007179—transforming growth factor beta receptor signaling pathway	3.53E-02	ID1, GDF5, HIPK2	
GO:0008584—male gonad development	3.68E-02	WNT5A, GATA6, ESR1	
GO:1902287—semaphorin-plexin signaling pathway involved in axon guidance	3.79E-02	PLXNA4, PLXNA2	
GO:0055119—relaxation of cardiac muscle	3.79E-02	RGS2, PDE5A	
GO:0007169—transmembrane receptor protein tyrosine kinase signaling pathway	3.82E-02	NTF3, FYN, NRG1	
GO:0001678—cellular glucose homeostasis	4.41E-02	FOXO1, PIK3R1	
GO:0060065—uterus development	4.41E-02	WNT5A, ESR1	
GO:0006636—unsaturated fatty acid biosynthetic process	4.72E-02	ELOVL2, SCD5	

Integrated analysis to identify crucial lncRNAs

By comparing the co-expression with ceRNA networks, five lncRNAs (SNHG9, LINC02202, UBAC2-AS1, PTCSC3 and myocardial infarction associated transcript (MIAT)) and 32 genes (such as PIK3R1, PTPRB) were found to be shared.

By comparing the hub genes enriched into KEGG pathways with the genes regulated by the above five lncRNAs (SNHG9, LINC02202, UBAC2-AS1, PTCSC3 and MIAT), we found the following ceRNA and co-expression axes may be important, including LINC02202 (upregulated)-hsa-miR-136-5p (downregulated)-PIK3R1 (upregulated), LINC02202 (upregulated)-hsa-miR-381-3p (downregulated)-FOXO1 (upregulated), MIAT (downregulated)-hsa-miR-18a-5p (upregulated)-ESR1 (downregulated) and LINC02202 (downregulated)-PIK3R1(downregulated). Furthermore, the comparison between hub genes enriched into KEGG pathways and the shared genes in two networks also indicated PTPRB related co-expression axis (LINC01119 (downregulated)-PTPRB (downregulated)) was also crucial.

Discussion

In present study, we identified three crucial lncRNAs (MIAT, LINC02202 and LINC01119) for adipogenesis from human ASCs. MIAT may sponge hsa-miR-18a-5p and influence the inhibition of hsa-miR-18a-5p on the expression of ESR1. LINC02202 may function as a ceRNA for hsa-miR-136-5p/hsa-miR-381-3p to respectively regulate the expressions of PIK3R1 and FOXO1; LINC02202 also may directly affect the transcription of PIK3R1. LINC01119 may co-express with PTPRB to impact its transcription. Although all these relationship pairs may be potentially important, LINC01119–PTPRB co-expression axis may be especially verifiable because their expression significance met the criterion of adjusted p-value < 0.05.

Although there have studies to show the roles of lncRNA MIAT for stem differentiation, only osteogenic (Jin et al., 2017) and endothelial cell (Wang et al., 2018) differentiation were investigated, without evidence to prove its effect on adipogenesis of human ASCs. A recent study revealed MIAT was an estrogen-inducible lncRNA and its expression was positively related to estrogen receptor (Li et al., 2018b). There was accumulating evidence to reveal that exposure of bone marrow stem cells to icariin or flavonoids of Herba Epimedii inhibited adipogenic differentiation, exhibiting decreased adipocyte numbers and downregulated mRNA expression of adipogenic differentiation markers, peroxisome proliferator‑activated receptor gamma (PPARγ) and CCAAT/enhancer‑binding protein α (C/EBPα) (Li et al., 2018c; Zhang et al., 2015); while treatment of bone marrow stem cells with estrogen receptor antagonist ICI182780 revered the effects of Herba Epimedii ingredient and promoted adipogenesis (Li et al., 2018c; Zhang et al., 2015). The study of Ihunnah et al. (2014) also demonstrated activation of estrogen receptor in ASCs inhibited adipogenesis by decreasing the recruitment of the adipogenic PPARγ onto its target gene promoters, whereas the use of estrogen receptor antagonism ICI 182780 or knockdown of estrogen receptor-α via lentiviral shRNA enhanced adipogenesis by increasing the expression of PPARγ. Thus, it can be hypothesized that MIAT may be lower expressed in adipogenic differentiation cells like ESR1, which was also confirmed in our study. However, the interaction mechanisms between MIAT and estrogen receptor remain unclear. In present study, we predicted that downregulated MIAT may be insufficient to sponge hsa-miR-18a-5p and lead to more hsa-miR-18a-5p to bind with the 3′ untranslated region of ESR1, inducing the lower expression of ESR1. This hypothesis may be indirectly demonstrated by the fact that miR-18a mimic significantly promoted mesenchymal stem cell (MSC) adipogenic differentiation, while the addition of miR-18a inhibitor obtained the negative effects on adipogenic differentiation of MSCs (Li et al., 2018a). The negative regulatory relationship between ESR1 and miR-18a were also validated in human trophoblast cell line by the luciferase assay (Zhu et al., 2015).

LINC02202 may be a newly identified lncRNA associated with stem cell differentiation because its role had not been previously mentioned in the literatures. In this study, we predicted upregulated LINC02202 may be involved in ASCs adipogenic differentiation by regulating phosphatidylinositol 3-kinase (PI3K) signaling. It has been reported that PI3K signaling pathway was strongly activated in MSCs under the adipogenesis-inducing hormone cocktail (Kim et al., 2017), and the addition of PI3K specific inhibitor LY294002 severely suppressed lipid accumulation, as well as the expression of adipogenic markers PPARγ and C/EBPα (Yu et al., 2008). PIK3R1 is a critical component of the PI3K signaling pathway and its expression was also demonstrated to be increased after the induction of adipocyte differentiation from preadipocytes 3T3-L1 (Kim et al., 2014a). Thus, theoretically, PIK3R1 may be upregulated in adipogenic differentiation cells compared with undifferentiated human ASCs, which was confirmed in our study. Activated PI3K/AKT signaling may promote adipogenesis through upregulating downstream transcription factors, such as FoxO1 (Yi et al., 2018) which may subsequently enhance the transcription of its target genes, PPAR-γ and C/EBP-α (Ambele et al., 2016; Munekata & Sakamoto, 2009); whereas persistent inhibition of FoxO1 with its antagonist AS1842856 (Zou et al., 2014) or knockdown of FoxO1 (Sun et al., 2017) was also observed to almost completely suppress adipocyte differentiation and lipogenesis. As expected, we also found FoxO1 was significantly high expressed during adipogenic differentiation. In addition to directly affect the transcription of PIK3R1, LINC02202 may function as a ceRNA for miR-136-5p and hsa-miR-381-3p to regulate the expression of PIK3R1a and FoxO1, respectively. Although there was no study to demonstrate these ceRNA interaction axes, the negative correlation between the expression of miR-136 and adipogenic markers C/EBPα and PPARα in subcutaneous adipose tissue of lambs may indirectly illuminate the importance of miR-136 for adipogenic differentiation (Meale et al., 2014). As expected, we also found miR-136-5p was significantly downregulated in adipogenic differentiation cells.

There was only one sequencing study to identify that LINC01119 was downregulated in colorectal cancer cells after hypoxia treatment (Han et al., 2019). Several authors had demonstrated hypoxia exposure was effective to enhance adipocyte differentiation from ASCs (Fink et al., 2004; Valorani et al., 2012; Kim et al., 2014b), which was medicated by the generation of reactive oxygen species (ROS) and activation of PI3K/Akt/mTOR (Kim et al., 2014b); the addition of ROS scavenger or Akt/mTOR inhibitor prevented adipocyte differentiation (Kim et al., 2014b). Thus, LINC01119 may have anti-adipose differentiation potential and lower expressed in adipogenic differentiation cells compared with undifferentiated human ASCs, which was validated in our study. However, its mechanisms for adipocyte differentiation remain unclear. We predicted LINC01119 may co-express with PTPRB. The study of Kim et al. showed ectopic over-expression of PTPRB inhibited the expression of adipocyte-related genes (such as PPAR-γ) and led to a reduced adipocyte differentiation from preadipocytes. Also, PTPRB was reported to suppress the tyrosine phosphorylation of VEGFR2 during adipocyte differentiation (Kim et al., 2019). Generally, VEGF functions by binding with VEGFR2, while transfection of VEGF to ASCs increased fat cell survival (Zhang et al., 2017). These findings suggest PTPRB may also be downregulated to promote VEGF secretion and activate its mediated pathways, ultimately inducing adipogenic differentiation from ASCs. This hypothesis was in line with our study showing PTPRB was lower expressed in adipocyte differentiation cells and was involved in angiogenesis.

There are some limitations in this study. First, only two datasets were submitted between 5 years until now, and not all were used for this analysis, which may cause some bias in results due to the small sample size and different data platforms. However, we believe the sequencing or microarray technology may be more mature recently and thus the results may be more believable. This was also indirectly reflected by the less overlapped genes if the other datasets were used (only two comparing GSE72429 with GSE25715; Guo & Cao, 2019) and thus, we renounced the use of multiple datasets and only the newly one. Moreover, this work investigated lncRNA co-expression and ceRNA mechanisms, which required the lncRNA and mRNA should be simultaneously analyzed. Thus, some datasets that only independently investigated lncRNA or mRNA were also excluded. Second, the crucial co-expression and ceRNA axes were obtained by database prediction, which may lead to many false positives. Therefore, further in vitro wet experiments (PCR, luciferase assay, knockdown or overexpression) are still indispensable to confirm the interaction between lncRNAs and miRNAs, lncRNA and mRNAs as well as the miRNAs and mRNAs and their roles during adipogenic differentiation of ASCs.

Conclusion

The present study preliminarily identified three new targets (lncRNA MIAT, LINC02202 and LINC01119) for inducing adipogenesis from human ASCs and promoting facial soft tissue reconstruction. They may be involved in adipogenesis by acting as a ceRNA (LINC02202-miR-136-5p-PIK3R1, LINC02202-miR-381-3p-FOXO1 and MIAT-miR-18a-5p-ESR1) or co-expressing with its targets (LINC02202-PIK3R1, LINC01119-PTPRB).

Supplemental Information

Supplemental Information 1 Raw data.

Click here for additional data file.

Supplemental Information 2 All differentially expressed genes.

Click here for additional data file.

Supplemental Information 3 LncRNA-mRNA co-expression pairs.

Click here for additional data file.

Supplemental Information 4 lncRNA–miRNA–mRNA interaction relationships.

Click here for additional data file.

Supplemental Information 5 GO enrichment results of PPI network genes.

Click here for additional data file.

Additional Information and Declarations

Competing Interests

Author Contributions

Data Availability

The authors declare that they have no competing interests.

Kana Chen conceived and designed the experiments, performed the experiments, analyzed the data, contributed reagents/materials/analysis tools, prepared figures and/or tables, authored or reviewed drafts of the paper, approved the final draft.

Shujie Xie analyzed the data, prepared figures and/or tables, approved the final draft.

Wujun Jin conceived and designed the experiments, authored or reviewed drafts of the paper, approved the final draft.

The following information was supplied regarding data availability:

Raw data is available in Supplemental Files.

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
