# Peer review of "Crucial lncRNAs associated with adipocyte differentiation from human adipose-derived stem cells based on co-expression and ceRNA network analyses"

_PeerJ, doi:10.7717/peerj.7544_

## Round 0.1 · original submission · Major Revisions

Your manuscript has been reviewed by three experts in the field. As you can find in their comments below, one of them only points out minor points while the others raise rather fundamental criticisms. Particularly, the comment that this work is based on relatively unreliable predictions should be seriously addressed. In addition, two reviewers mention about the positive/negative sides of using of heterogeneous data (microarray/RMA-seq). Please read their comments carefully and revise the manuscript accordingly. Looking forward to your revised manuscript.

Reviewer 1 ·

Basic reporting

no comment

Experimental design

Experimental design is well designed .

Validity of the findings

The finding is novel.

Additional comments

Thank you for kindly give me the chance to reviewer this manuscript. I have carefully read and review the manuscript" Crucial lncRNAs associated with adipocyte differentiation from human adipose-derived stem cells based on co-expression and ceRNA network analyses" The manuscript described a study which tried to screen crucial lncRNAs associated with adipogenic differentiation of HASCs based on the coexpression and ceRNA mechanisms. Overall, the analysis workflow is correct and the manuscript is well written. We only suggested to giving minor modification to consider to accept this paper.
line 168: which contained 440 positive connections (Figure 2; (Supplemental Information 3). Please delete the Repeated parenthesis.
Line 177: which was used for constructing the ceRNA network. Were not was ?
Line 209: ESR1(downregulated). Furthermore…a space should be deleted before Furthermore.
There were two panels for figure 2,3, please use A and B to display them.

Reviewer 2 ·

Basic reporting

no comment

Experimental design

The authors used microarray data. In addition to these data, if the authors include RNA-seq data in their analysis, the biological finding present in this paper will be more reliable. Are there any RNA-seq data which can be utilized in the authors' studies?

Validity of the findings

The reliability of the authors' finding is questionable because the authors' analyses include the following two drawbacks.
(1) The authors used p-value for selecting differentially expressed genes/lncRNAs/miRNAs; However, they should use adjusted p-values reported in e.g., Supplementary Table S2. Additionally, the authors should provide adjusted p-values for lncRNA and miRNA (Table 1 includes only partial information).
(2) It is well known that miRNA target predictions lead to many false positives. Another criticism of this study is to depend greatly on the predictions. I suggest the authors use other resources for ceRNAs (e.g., [http://starbase.sysu.edu.cn/fstarbase2/mrnaCeRNA.php](http://starbase.sysu.edu.cn/fstarbase2/mrnaCeRNA.php) ) in their analyses.

Reviewer 3 ·

Basic reporting

No comments. The manuscript is well written. Literature references are sufficient and relevant to the hypothesis.

Experimental design

One major comment is related to the choose of the reference data. Why were this two assays selected? This should be explained in the manuscript, mainly considering that expression data came from different strategies (microarray and RNA-Seq).

Validity of the findings

Though analysis of data is well done and results may be interesting the overall manuscript seems to contain preliminary data. The major findings reported appear as highly speculative and must be confirmed experimentally.

---

## Round 0.2 · accepted · Accept

Your revised manuscript has been re-reviewed by the same three referees. All of then now recommends its acceptance and thus I am happy to inform you that I will accept this manuscript for publication in PeerJ. Congratulations!

Reviewer 1 ·

Basic reporting

no comment

Experimental design

no comment

Validity of the findings

no comment

Additional comments

no comments

Reviewer 2 ·

Basic reporting

no comment

Experimental design

no comment

Validity of the findings

no comment

Additional comments

The authors addressed most of my concerns.

Reviewer 3 ·

Basic reporting

no comment

Experimental design

The authors have answered the major comments about the experimental design.

Validity of the findings

Results are highly speculative, the authors have stated this in the manuscript

Additional comments

The author have answered all major comments. Though still have doubts about how meaningful are these results, the manuscript can be published in Peer J.